# ADAPTIVE CONVOLUTIONAL NEURAL NETWORKS

## ABSTRACT

The quest for increased visual recognition performance has led to the development of highly complex neural networks with very deep topologies. To avoid high computing resource requirements of such complex networks and to enable operation on devices with limited resources, this paper introduces adaptive kernels for convolutional layers. Motivated by the non-linear perception response in human visual cells, the input image is used to define the weights of a dynamic kernel called Adaptive kernel. This new adaptive kernel is used to perform a second convolution of the input image generating the output pixel. Adaptive kernels enable accurate recognition with lower memory requirements; This is accomplished through reducing the number of kernels and the number of layers needed in the typical CNN configuration, in addition to reducing the memory used, increasing 2X the training speed and the number of activation function evaluations. Our experiments show a reduction of 66X in the memory used for MNIST, maintaining 99% accuracy and 16X memory reduction for CIFAR10 with 92.5% accuracy.

## 1 INTRODUCTION

Convolutional Neural Networks (CNN) have demonstrated their capacity to achieve state of the art accuracy in image classification, semantic segmentation, and object detection. In order to increase the accuracy of recognition, most of these works rely on deeper and deeper architectures with millions of parameters. The reason behind this is that more complex features can be abstracted as we add more layers to a network. Unfortunately such models can not be used in embedded devices, cellphones or drones due to the size of the models. Recently there are many approaches like ShuffleNet Xiangyu Zhang (2017), MobileNet Andrew G. Howard (2017), HENet Qiuyu Zhu (2018) and SqueezeNet Forrest N. Iandola (2017) trying to generate small models with an small drop in accuracy. The proposed method generates the smallest model with comparable accuracy. In addition this method can be combined with most of the existent techniques like ResNet generating an improved version of it.

Previous research has demonstrated that the response of visual cells is a non-linear function of their stimuli (Szulborski & Palmer, 1990). Thus, finding non-linear models that best represent data is imperative. Given that a typical convolutional layer is a linear system, its ability to express this response is limited by the layers and the number of neurons in the intermediate layers. The use of a non-linear neuron, as it was firstly explored to solve the XOR problem by Minsky & Papert (1969), (which cannot be solved by a first order neuron, but it can by a second order neuron), seems an appropriate way to tackle this kind of non linearity issue. Ideally, such non-linear approaches should be able to provide similar accuracy as compared to traditional CNNs, albeit at a much lower computation and memory costs. In other hand, in computer vision the filter used to extract borders is different to the filter used to extract corners, etc. Our method uses the input image to define the filter that better fits in on the specific location.

Motivated by this, we developed a convolutional kernel, that includes non-linear transformations obtaining similar results as the state of the art algorithms, while yielding a reduction in required memory up to 14x in the CIFAR10 (Krizhevsky, 2009) classification, and up to 66x for the MNIST classification. The main contributions of this paper are: (i) we present the non-linear convolutions designed for high visual classification accuracy under memory constraints; (ii) using the proposed convolutions, we present a network design that is partially pre-defined and is capable of completing self-definition during the pattern evaluation phase, including defining the convolutional kernels on the fly depending of the input pattern; (iii) we present a method to tackle problems associated with

higher order neural networks like the saturation of the activation due to the N-order of multiplications used; (iv) a method to constrain every new dynamically generated weight to a pre-known range defined by the activation function, for instance if hyperbolic tangent function was used, the dynamically generated weights would be all in $(-1, 1)$ range; and (v) a pytorch-based implementation located in https://github.com/adapconv/adaptive-cnn (2018).

This work is organized as follows: Section 2 describes previous approaches, Section 3 explains in detail our proposed method, Section 4 shows the results for MNIST, CIFAR10 and Navigation. We end with our conclusions in Section 5.

## 2 RELATED WORK

Different ways of increasing the accuracy of neural networks have been addressed in the existing scientific literature, most of these rely on the use CNNs, as they are generalized linear models, and their level of abstraction is low (Lin et al., 2013). Some approaches have dealt with this low abstraction by having additional layers (Krizhevsky et al., 2012; Simonyan & Zisserman, 2014), resulting in a considerable increase in the accuracy on different datasets, e.g. CIFAR10 (Krizhevsky, 2009) and ImageNet (Deng et al., 2009). However, although the network depth has shown a crucial importance in neural network performance, the difficulty to train the network also increases, and moreover, the accuracy of the network drops (He et al., 2015).

A proposal to tackle this problem is the use of ResNet blocks He et al. (2015), which given a set of inputs $X$, with an associated label $Y$ and a function $H(x)$ that maps $X$ to $Y$, the networks define a building block $y = F(x) + x$, where $F(x)$ represents the residual mapping to be learned. This residual network allows to train a deeper network without being affected by the degradation problem, using a larger amount of layers and parameters.

In the work presented in Lin et al. (2013) a nonlinear function approximator is proposed as a solution to increase the level of abstraction. The typical convolutional kernel is replaced with a micro network, i.e. a non linear approximator. A multilayer perceptron is used as the instantiation of this micro network, and by sliding the micro network over the input in a similar manner as convolutional neural networks the feature maps are obtained.

In the approach of Brabandere et al. (2016), a dynamic filter module is introduced, where the filters used for the convolution are generated dynamically depending on the input. This module consists of two parts: a filter generating network, which generates sample specific parameters given an input; and the dynamic filtering layer, which applies the parameters to another input. Although the mentioned work presents similar components to our proposal, it is important to notice the differences between these: in Brabandere et al. (2016) a CNN is trained to get the convolutional kernels, and then another network is defined to generate such kernels, where this second network is trained independently. In our work, we define a second order convolutional kernel trained using a novel training rule, which is explained in detail in the next section.

## 3 METHOD

An Adaptive Kernel is a dynamic kernel that changes its weights by itself depending on the input image. An Adaptive Kernel can be viewed as an array of traditional kernels. For instance, each element in a 3x3 adaptive kernel is a 3x3 linear kernel, as shown in the Figure 1.

The convolution of each kernel element $Q_{u,v}$ with the window $X$ of the input image generates a component of the new kernel $K_{u,v}$ after applying the activation function (Figure 1). This new dynamically generated kernel $K$ is convolved again with the same window $X$ of the input image (Figure 2).

As result of this second order convolution and the activation function an output pixel value is obtained. By sliding the window through all the input image, we generate the final filtered image.

### 3.1 FEED FORWARD

Convolving the kernel $Q_{u,v}$ in the Figure 1 and using $tanh$ as activation function the weights $K_{u,v}$ are generated

$$K_{u,v}(\sigma) = tanh(\sigma_{u,v}), \quad (1)$$

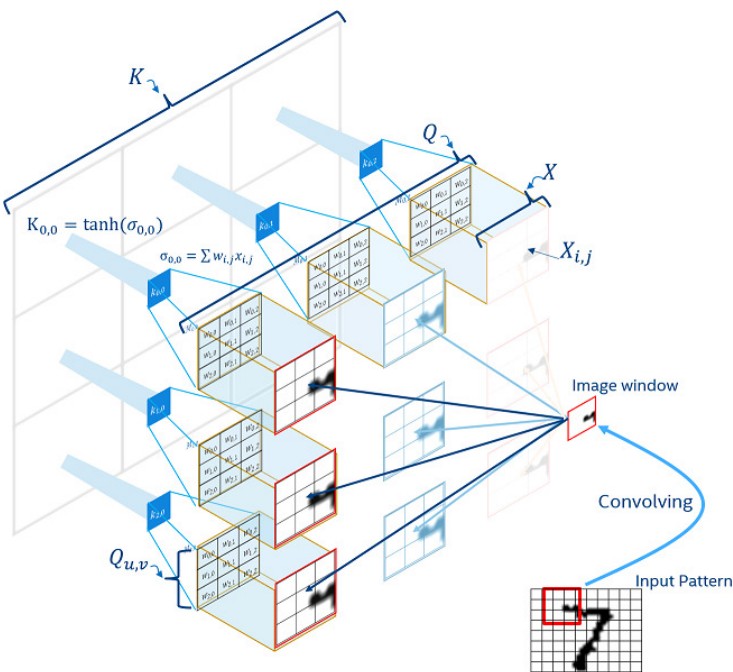

Figure 1: One adaptive kernel created by the convolution of the input image with a matrix of kernels.

where $\sigma_{u,v}$ is the convolution of each linear kernel $Q_{u,v}$ with the input image

$$\sigma_{u,v} = \sum_{i=0}^{N-1} \sum_{j=0}^{N-1} Q_{(u,v)_{i,j}} x_{i,j}, \tag{2}$$

This new kernel $K_{u,v}$ is now convolved with the input image to compute $S$ in this location

$$S = \sum_{u,v} x_{u,v} K \left( \sum_{i,j} Q_{u,v_{i,j}} x_{i,j} \right) \tag{3}$$

and finally the output pixel is computed using hyperbolic tangent as activation function like $f = tanh(S)$. Another activation functions can be used, like sigmoid or Relu. We use hyperbolic tangent to generate weights in the range of $(-1, 1)$.

## 3.2 TRAINING

For the Adaptive kernel a new training rule is obtained using gradient descent technique in order to adjust the weights of adaptive kernel $Q$. The kernel can be seen as an array of $NxN$ traditional kernels, using $Q_{u,v}$ to refer to each linear kernel and using $(i, j)$ to refer the elements (weights) of each $Q_{u,v}$ kernel. It means the element $Q_{(u,v)_{(i,j)}}$ is a scalar value and represents a weight.

The kernel output $f(S)_{k,l}$, where $f$ is the activation function and $S$ represents sum of input $x_{u,v}$ weighted by $\sigma_{u,v}$ as shown in Equation (3). The error for output pixel $(k, l)$ denoted as $E_{k,l}$ is given by:

$$E_{k,l} = \frac{1}{2}(d_{k,l} - f(s)_{k,l})^2, \tag{4}$$

where $d_{k,l}$ represents the expected output at coordinates $(k, l)$. The training rule for the $t - th$ iteration $(Q^t)$ is given by the error derivative per component $(u, v)_{i,j}$:

$$\frac{\partial E_{k,l}}{\partial Q^t_{(u,v)_{(i,j)}}} = (d_{k,l} - f(s)_{k,l}) \frac{\partial f(s)_{k,l}}{\partial s_{k,l}} \frac{\partial S_{k,l}}{\partial Q^t_{(u,v)_{(i,j)}}} \tag{5}$$

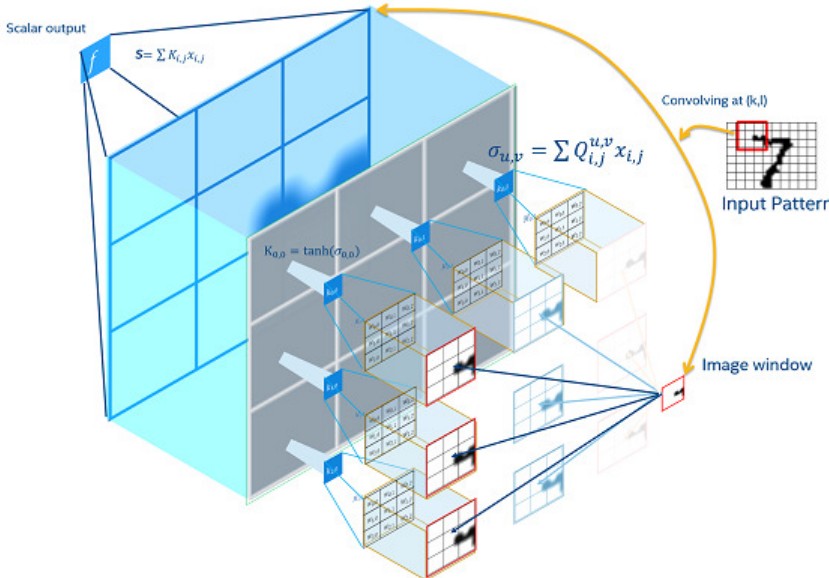

Figure 2: A single resulting pixel output from the convolution of the input image and the high order kernel that generates a dynamic kernel which is convolved with the input image again.

The gradient descent updates of the weights $Q^t_{(u,v)_{i,j}}$ are given by:

$$Q^{t+1}_{(u,v)(i,j)} = Q^t_{(u,v)(i,j)} + \gamma \sum_{k,l} (d_{k,l} - f(s)_{k,l}) \frac{\partial f(s)_{k,l}}{\partial Q^t_{(u,v)(i,j)}} \tag{6}$$

where $\gamma$ represents the learning rate. Using the hyperbolic tangent as the activation function $f(S)_{k,l} = \tanh(S_{k,l})$:

$$\frac{\partial f(S)_{k,l}}{\partial S_{k,l}} \frac{\partial S_{k,l}}{\partial Q^t_{(u,v)(i,j)}} = (1 - f(S)^2_{k,l}) \frac{\partial S}{\partial Q^t_{(u,v)(i,j)}}, \tag{7}$$

where $S_{k,l}$ is the weighted sum computed as:

$$S_{k,l} = \sum_{u=0}^{N-1} \sum_{v=0}^{N-1} (K^{k,l}_{u,v})(x_{u+k,v+l}). \tag{8}$$

Here, $N$ is the size of the kernel, and $K^{k,l}_{u,v}(\sigma) = \tanh(\sigma^{k,l}_{u,v})$, with:

$$\sigma^{k,l}_{u,v} = \sum_{i=0}^{N-1} \sum_{j=0}^{N-1} Q_{(u,v)_{i,j}} x_{i+k,j+l}, \tag{9}$$

Since the hyperbolic tangent was used as the activation function, the gradient is:

$$\frac{\partial f(s)_{k,l}}{\partial Q^t_{(u,v)(i,j)}} = (1 - f(S)^2_{k,l}) \sum_{u=0}^{N-1} \sum_{v=0}^{N-1} (1 - (K^{k,l}_{u,v}(\sigma))^2)(x_{u+k,v+l})(x_{i+k,j+l}) \tag{10}$$

by replacing (10) in (6), the final training rule is defined by:

$$Q^{t+1}_{(u,v)(i,j)} = Q^t_{(u,v)(i,j)} + \gamma \sum_{k,l} (d_{k,l} - f(S)_{k,l})(1 - f(S)^2_{k,l})\delta_{k,l} \tag{11}$$

where $\gamma$ is the learning rate, $d_{k,l}$ is the desired value at the window position $(k, l)$, $f$ is the activation function and $\delta_{k,l}$ is

$$\delta_{k,l} = \sum_{u,v} (1 - (K^{k,l}_{u,v}(\sigma))^2)(x_{u+k,v+l})(x_{i+k,j+l}) \tag{12}$$

For this training rule $tanh$ was used as activation function to generate the kernel weights, it ensures all weight are in the range of $(-1, 1)$. Additionally, this layer allows the reduction of filters in the subsequent layers without affecting the performance of the network, on the experimental results different experiments will be used to compared against state of the art for MNIST and CIFAR10 to show a significant memory compression using the proposed models.

## 4 RESULTS

Our implementation of the adaptive kernels was written as a layer of Caffe (Jia et al., 2014) with forward and backward propagation. For CIFAR-10 (Krizhevsky, 2009) we use Nesterov (Nesterov, 1983) as training rule, with momentum set to 0.9, initializing the learning rate to 0.1, dropping it by a factor of 5 every 20 epochs, and a weight decay of 0.0001. We use the same weight initialization as He et al. (2015). A 32x32 crop is randomly sampled from a 40x40 image or its horizontal flip, with the per-pixel mean subtracted and divided by the channel standard deviation. For MNIST we use the same parameters, changing the learning rate to 0.01 and the weight decay to 0.0005. With no data augmentation or pre-processing.

### 4.1 EXPERIMENT 1: MNIST

The MNIST is a public data set, it consists of 60,000 28x28 grey scale images in 10 classes (hand-written numbers), with 6000 images per class. There are 50,000 training images and 10,000 test images in the official data. Here our approach has three main advantages: memory reduction, increment of accuracy outperforming the traditional model of CNNs, and the learning speedup. The results produced by the implementation of the MNIST digits recognition, show a big memory compression, using 66X less memory measured through parameter reduction, additionally high accuracy was achieved 2x faster. In the table 1a the LeNet neural network architecture used as reference is detailed[1] and table 1b describes our topology.

Table 1: Topology comparison for MNIST.

(a) LeNet CNN Topology as in tutorial

| Layer | Units | Type |
|---|---|---|
| Layer1 | 20 Kernels | Conv $5x5$ |
| Layer2 | 50 Kernels | Conv $5x5$ |
| Layer3 | 500 Neurons | FC |
| Layer4 | 10 Neurons | FC |

(b) Our Neural Network Topology

| Layer | Units | Type |
|---|---|---|
| Layer1 | 5 Kernels | Adaptive $5x5_{5x5}$ |
| Layer2 | 10 Kernels | Conv $5x5$ |
| Layer3 | 20 Neurons | FC |
| Layer4 | 10 Neurons | FC |

The (figure 3A) shows a few examples of how the Adaptive kernel is changing according to the input window. The Figure 3B shows all the kernels generated for random sample of digit seven, For all the back ground pixels the kernels is simple neutral it does not extract any features there.

There are many different NN Models of MNIST classification, In the table 2 a subset is presented, selecting only the models having over 99% accuracy with small number of parameters.

Table 2: MNIST Accuracy vs Number of parameters for >99% accuracy

| Neural Network | Depth | #Parameters |
|---|---|---|
| LeNet (BAIR/BVLC, 2018) | 4 | 431K |
| LetNet5 Yann Lecun (1998) | 7 | 60K |
| 50-50-200-10NN MarcAurelio Ranzato (2006) | 4 | 226K |
| Best Practices (Patrice Y. Simard, 2003) | 4 | 132.5K |
| **Adaptive Kernels CNN** | 4 | **6.52K** |

---

[1]FC:Fully connected layer

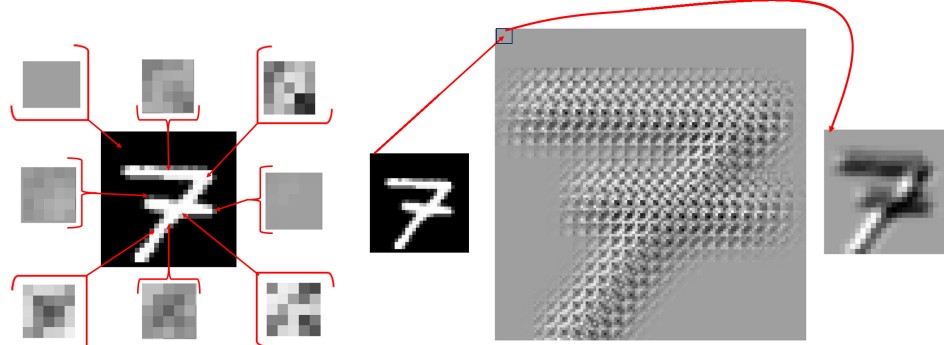

Figure 3: A) A single kernel generated in different positions of the input image. B)Every input window is convolved by a different filter generated on the fly using the input image.

In order to perform a Hyper-parameter sensibility analysis, eight different models with three layers were trained using MNIST. By increasing the number of kernels in the first layer, a saturation on the accuracy can be seen, but this saturation can be mitigated incrementing the number of kernels in the second layer (3).

Table 3: Hyper-parameter sensibility

| Layer type | M1 | M2 | M3 | M4 | M5 | M6 | M7 | M8 |
|---|---|---|---|---|---|---|---|---|
| Adaptive | 4 | 5 | 6 | 7 | 4 | 5 | 6 | 7 |
| Conv. | 10 | 10 | 10 | 10 | 20 | 20 | 20 | 20 |
| F.C. | 10 | 10 | 10 | 10 | 10 | 10 | 10 | 10 |
| Accuracy | 98.14 | 98.17 | 98.24 | 98.30 | 98.57 | 98.65 | 98.85 | 99.04 |
| #parameters | 6K | 6.8K | 7.7K | 8.6K | 9.5k | 10.6K | 11.7K | 12.8K |

In this scenario our technique is the smallest CNN model that reaches 99% without any pre-processing in only 5 epochs, in contrast, the LeNet reached 97% after 9 epochs. In terms of the number of operations the LeNet as in the tutorial has 2.29M MAC operations, while our method has 1.23M MAC operations for MNIST.

## 4.2 EXPERIMENT 2:CIFAR10

The CIFAR-10 is a public data set, it consists of 60,000 32x32 color images in 10 classes, with 6000 images per class. There are 50,000 training images and 10,000 test images in the official data. In this scenario CIFAR10 was used for testing and we used horizontal flipping, padding and 32x32 random cropping for data augmentation of the training dataset. In the selected topology: only the first layer uses adaptive kernels of $(3x3)_{(3x3)}$ in order to highlight the impact of a single layer in the full topology, being the first layer where the main feature extraction takes place. It has 8 convolutional layers and ends with ten outputs in the fully connected layer.

In order to compare against related work we include some of the latest topologies used for CIFAR10 pattern recognition problem. Our intention is not to outperform the accuracy of these topologies. Instead, the idea is to achieve similar results with a smaller model. This approach can potentially enable us to target embedded systems. In the Table 4 we provide a summary of the most popular methods recently used for CIFAR10. As the Table 4 shows the focus of the prior work has been the improvement of recognition accuracy without regard to the amount of memory and time required for classification.

The neural networks achieving the highest scores use over 20 million parameters, making it very hard to implement them in low capacity embedded devices. Our solution reduces memory usage; it only uses 200K parameters. While there is no simple way to determine the efficiency of a neural

Table 4: CIFAR10 Classification error vs Number of parameters

| Neural Network | Depth | #Parameters | Error% |
|---|---|---|---|
| All-CNN (Springenberg et al., 2014) | 9 | 1.3M | 7.25 |
| ResNet Stochastic Depth Huang et al. (2016) | 110 | 1.7M | 5.23 |
| Pre-act Resnet (He et al., 2016) | 1001 | 10.2M | 4.62 |
| Wide ResNet (Zagoruyko & Komodakis, 2016) | 40 | 55.8M | 3.8 |
| PyramidNet (Han et al., 2016) | 110 | 28.3M | 3.77 |
| Wide-DelugeNet (Kuen et al., 2016) | 146 | 20.2M | 3.76 |
| Steerable CNN (Cohen & Welling, 2016) | 14 | 9.1M | 3.65 |
| ResNet Xt (Xie et al., 2016) | 29 | 68.1M | 3.58 |
| Wide ResNet with Singular Value Bounding (Jia, 2016) | 28 | 36.5M | 3.52 |
| Oriented Response Net (Zhou et al., 2017) | 28 | 18.4M | 3.52 |
| Baseline wide ResNet | 28 | 36.6M | 3.62 |
| Volterra-based Wide ResNet (Zoumpourlis et al., 2017) | 28 | 36.7M | 3.51 |
| MobileNetV1 (Andrew G. Howard, 2017) | 28 | 3.2M | 10.76 |
| MobileNetV2 (Mark Sandler, 2018) | 54 | 2.24M | 7.22 |
| shuffleNet 8 Groups (Xiangyu Zhang, 2017) | 10 | 0.91M | 7.71 |
| shuffleNet 1 Group (Xiangyu Zhang, 2017) | 10 | 0.24M | 8.56 |
| HENet (Qiuyu Zhu, 2018) | 9 | 0.7M | 10.16 |
| VGG Karen Simonyan (2015) | 14 | 14M | 7.36 |
| **Adaptive Kernels CNN** | 10 | **0.2M** | 7.48 |

network, our target is the highest accuracy with the least amount of memory as shows (Table 4). Based on accuracy the Volterra-based Wide ResNet Zoumpourlis et al. (2017) provides the best score but uses 36M parameters. Our solution, the one with less number of parameters represents a compression of 184X with 4% drop in accuracy or comparing with All-CNN Springenberg et al. (2014) our solution represents a 6X memory reduction with $0.2\%$ accuracy drop.

Additional experiments were performed taking as base the ResNet18 topology, in this experiment the number of parameters was not taken into account and only the first layer was changed to use 16 adaptive kernels instead of 64 convolutional kernels, as in the typical topology, in order to show the contribution of one adaptive layer. Although the topology with an adaptive layer has less feature maps in the first layer, it can achieve even better results than ResNet50 see table 5.

Table 5: Combining Adaptive layers with ResNet

| Neural Network | Depth | #Parameters | Error% |
|---|---|---|---|
| ResNet18 Kaiming He (2015) | 18 | 11M | 6.98 |
| ResNet50 Kaiming He (2015) | 50 | 25.6M | 6.38 |
| ResNet100 Kaiming He (2015) | 100 | 44.5M | 6.25 |
| **Adaptive + ResNet18** | 18 | **10.8M** | 6.33 |

## 4.3 EXPERIMENT 3:GENERALIZATION

In this experiment an internal data set was used, in order to train a neural network model, that drives a robot to navigate inside of a known room. Basically given an input image $I$ from the robot camera, estimate a required direction $\alpha \in [0, 360]$ and a distance $d \in [0, 100]$, that drives the drone to reach the center $c(x_c, y_c)$ of a known region. For this purpose, a NN architecture was created inspired by DroNet Loquercio et al. (2018), instead of a standard convolutional layer, an adaptive convolutional layer was used (Table 6b). The last two layers of the network are trained considering a classification problem, in order to estimate a steering angle class and a distance class.

The compression is achieved by reducing the number of filters in the three ResNet blocks. In the table we have $k$, $2k$, and $4k$ kernels for each block respectively. When the compression increases

Table 6: Topology comparison for generalization.

(a) DroNet Neural Network Topology

| Layer | Units | Type |
| --- | --- | --- |
| Layer1 | 32 Kernels | Conv $5x5$ |
| Layer2-5 | k-Kernels | ResNet 5,3,1 |
| Layer6-8 | 2k-Kernels | ResNet 5,3,1 |
| Layer9-11 | 4k-Kernels | ResNet 5,3,1 |
| Layer12 | 120 Neurons | FC |
| Layer12' | 100 Neurons | FC |

(b) Our Neural Network Topology

| Layer | Units | Type |
| --- | --- | --- |
| Layer1 | 5 Kernels | Adaptive $5x5_{5x5}$ |
| Layer2-5 | k-Kernels | ResNet 5,3,1 |
| Layer6-8 | 2k-Kernels | ResNet 5,3,1 |
| Layer9-11 | 4k-Kernels | ResNet 5,3,1 |
| Layer12 | 120 Neurons | FC |
| Layer12' | 100 Neurons | FC |

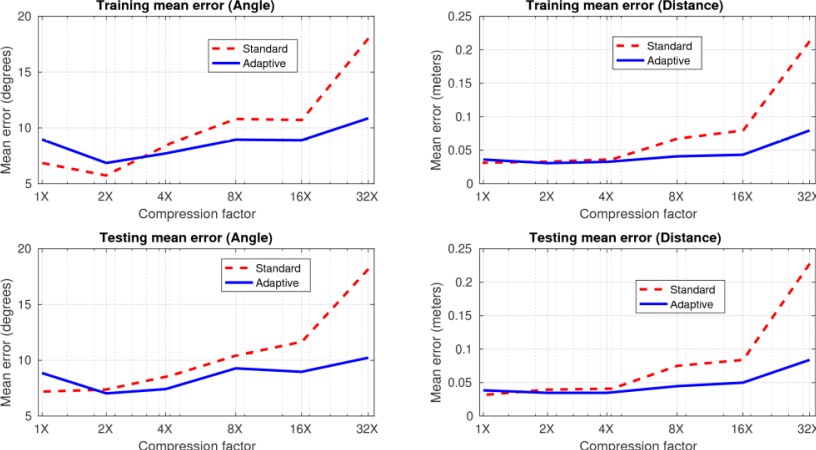

Figure 4: Accuracy vs parameter reduction with standard and adaptive models, training (top) and testing (bottom).

the accuracy drops, but the Adaptive kernels helps to keep a better accuracy in comparison with the traditional convolutional layers (4).

## 4.4 CONCLUSION

In this paper we presented adaptive convolutional kernels, capable of redefining the convolutional kernel during the inference time, depending on the input image. Our technique not only reduces memory usage, but also reduces the training time. Additionally, our results show adaptive convolutional kernels generalize better than traditional CNNs. Kernels adapt dynamically to extract better features depending of the input image, and in terms of efficiency, our method has shown to be very compelling generating 10X lighter solutions with less than $1\%$ accuracy drop. Our solution should be able to provide direct impact to the computational cost of the inference on embedded systems, increasing the operational scope of these systems. In terms of accuracy, our proposed Adaptive kernel can imitate any traditional constant kernel (by training the dynamic kernel to be constant independent of the input); this means that it can be trained to generate the weights in the best in class solution and start from there to outperform it. As in traditional CNNs, the increment of the number of kernels in a layer produces some saturation, with a marginal increment of accuracy, thus the topology also plays an important role. In our experiments it was observed that less adaptive kernels in a layer generates comparable or even better level of abstraction than a higher number of traditional convolutional kernels in a layer. For instance, in ResNet18, the 64 convolutional filters in the first layer were changed to only 16 Adaptive filters producing better results than the topology with 50 layers. In addition fully connected layers can use this technique, one adaptive layer can replace two traditional layers.

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
