# OpenReview forum: "Adaptive Convolutional Neural Networks"
_ICLR.cc/2019/Conference_

### Official Review · AnonReviewer3 · 2018-11-01
**Review of Adaptive Convolutional Neural Networks.**

**Rating:** 4
**Confidence:** 4

**Review:**

The paper introduces adaptive kernels (that adapts its weights as a function of image content) to the framework of CNN. The benefit of adaptive kernels is the reduction of memory usage (at training and at the inference time) as well as training speedups (up to 2x). The kernels are evaluated on two datasets MNIST and CIFAR10

I like the idea of building models that are memory efficient at training and at evaluation time. However, the evaluation of the proposed adaptive kernels is rather limited. In order to improve the paper, the authors could take into consideration the following points:

1. Why there is still a need to combine adaptive convolutions with regular convolutions? What would the model performance be for a model with only adaptive kernels?
2. I might have missed it, but I couldn't find any motivation on why tanh is used as nonlinearity. Would the method work with relu?
3. Traditional convolutional kernels together with max pooling operations ensures some degree of translation invariance. How big is the generalization gap for the tested models when adaptive kernel is used?
4. How sensitive are the results to the number of adaptive kernels in the layers.
5. Adaptive kernels have only been tested in the first convolutional layer, would the adaptive kernels work well also in different layers?
6. On CIFAR10 the results seem to be worse that other methods. However, it is important to note that the Adaptive Kernels CNN has way less parameters. It would be interesting to see how the performance of adaptive kernels based CNNs scales with the number of parameters.
7. The evaluation on two datasets seems to be rather limited, additional comparisons should be included.
8. The authors acknowledge the similarities (and some differences) with Brabandere et al (2016). It might be beneficial to include comparison to this approach in the experimental section. Moreover, given the similarities, it might be good to discuss the differences in the approaches in the introduction section.
9. The ideas presented in the paper seems related to general concept of hypernetworks, where one network learns (or helps to learn) paramenters of the other network. It would be nice to position the ideas from the paper w.r.t. this line of research too.
10. Another related paper seems to be Spatial Transformer Networks (Jaderberg et al.).

I like the drawings, however, the font on the drawings is too small - making it hard to read.

Some typos:
1. the difficult to train the network
2. table 2: Dynamic -> Adaptive?

Overall, the paper presents interesting ideas with some degree of originality. I'd encourage the authors to extend the intro and position the ideas w.r.t. existing works and extend the evaluation.

---

> ### Author Response · Authors · 2018-11-21
> **Thank you, very good review**
>
> 1. Why there is still a need to combine adaptive convolutions with regular convolutions? What would the model performance be for a model with only adaptive kernels?
>
> R)It is possible to change all the layers to use Adaptive convolutions, we replaced only one to measure the unitary contribution. We chose the first one because it is where the feature extraction is performed, in addition fully connected layers can use this technique. One Adaptive layer can replace two traditional layers.  (Added to the paper)
>
> 2. I might have missed it, but I couldn't find any motivation on why tanh is used as nonlinearity. Would the method work with relu?
>
> R) We can use any Activation function. We actually tested ReLu with good results, but we chose tanh because it generates weights in the range of (-1,1) avoiding large values given by ReLu. (Added to the paper)
>
> 3. Traditional convolutional kernels together with max pooling operations ensures some degree of translation invariance. How big is the generalization gap for the tested models when adaptive kernel is used?
> R)we added an experiment to test the generalization
>
> 4. How sensitive are the results to the number of adaptive kernels in the layers.
>
> R) As in traditional CNNs, the increment of the number of kernels in a layer produces some saturation, with a marginal increment of accuracy. In our experiments it was observed that 5 dynamic kernels generates comparable level of abstraction than 30 traditional convolutional kernels. (Added to the paper)
>
> 5. Adaptive kernels have only been tested in the first convolutional layer, would the adaptive kernels work well also in different layers?
> yes we have test several layers with adaptive kernels. but we focus on report the results on the first layer to highlight the contribution
>
> 6. On CIFAR10 the results seem to be worse that other methods. However, it is important to note that the Adaptive Kernels CNN has way less parameters. It would be interesting to see how the performance of adaptive kernels based CNNs scales with the number of parameters.
>
> R)We added a new experiment where we show how the performance of adaptive kernels improves with the increment of parameters. In order to make a fairer comparison we also added another experiment where we compare the accuracy of ResNet18 with 1 adaptive layer against ResNet18, ResNet50 and ResNet101 and it can be seen that the adaptive one performs even better than ResNet50.
>
> 7. The evaluation on two datasets seems to be rather limited, additional comparisons should be included.
> R) Added another experiment, where we use DroNet as base to show the benefit of combine Adaptive layers with ResNet, in this experiment we test different configurations to compress the network up to 32X  (Added to the paper)
>
> 8. The authors acknowledge the similarities (and some differences) with Brabandere et al (2016). It might be beneficial to include comparison to this approach in the experimental section. Moreover, given the similarities, it might be good to discuss the differences in the approaches in the introduction section.
>
> R)they train a NN to generate a model of another NN, we  have a ACNN that learns how to generate its filters.  (in the intro)
>
> Some typos:
> 1. the difficult to train the network
> 2. table 2: Dynamic -> Adaptive?
> Very Good Catch 
>
> Overall, the paper presents interesting ideas with some degree of originality. I'd encourage the authors to extend the intro and position the ideas w.r.t. existing works and extend the evaluation.
> We compared now against: mobileNet, ShuffleNet, HENet, SqueezeNet, we have less number of parameters or better accuracy or both (Added to the paper)

---

### Official Review · AnonReviewer1 · 2018-11-02
**Is this really a type of convolutional network?**

**Rating:** 4
**Confidence:** 3

**Review:**

The paper develops a new 'convolution' operation.
I think it is misleading to call it a convolution, as (a) it is not a convolution mathematically, and (b) fast convolution techniques (Fourier, Winograd) cannot be applied, so claims to greater efficiency may be misleading.

p2-3, Section 3.1 - I found the equations impossible to read. What are the subscripts over?
In (2) is (N+1)x(N+1) the kernel size (sums are over 0,1,...,N?)??
Is the output of the first convolution a single HxW feature planes, or a HxWx(N+1)x(N+1) tensor?

Equation (4). What is d_{k,l}? A pixel-wise target label? Where does it come from?

Experimental section: Like depthwise convolutions, you seem to achieve reasonable accuracy at fairly low computational cost. It would therefore be much more interesting to compare your networks with ShuffleNet style networks designed for computational efficiency, rather than networks designed mainly to push the benchmark numbers down whatever the cost.

It would be helpful to have the computational cost of the network in FLOPs, and running time compared a regular ConvNet using Winograd/Fourier convolutions.

---

> ### Author Response · Authors · 2018-11-21
> **Thank you, very good feedback**
>
> I think it is misleading to call it a convolution, as (a) it is not a convolution mathematically, and (b) fast convolution techniques (Fourier, Winograd) cannot be applied, so claims to greater efficiency may be misleading.
>
> R)We believe as future work our algorithm can be combined with Winograd techniques for optimization. For instance winograd is designed to use a batch of images to convolve with a kernel, here an image convolves with a “batch of kernels”. There is no reason why those two techniques can be merged. In our implementation we perform a set of convolutions with the input image where FFT can be applied too.
>
> p2-3, Section 3.1 - I found the equations impossible to read. What are the subscripts over?
> In (2) is (N+1)x(N+1) the kernel size (sums are over 0,1,...,N?)??
> Is the output of the first convolution a single HxW feature planes, or a HxWx(N+1)x(N+1) tensor?
>
> Very good Catch it should be (N)x(N) instead of (N+1)x(N+1). (Fixed on the paper)
> Equation (4). What is d_{k,l}? A pixel-wise target label? Where does it come from?
> {k,l} locate the convolving window inside of the input image
>
> Experimental section: Like depthwise convolutions, you seem to achieve reasonable accuracy at fairly low computational cost. It would therefore be much more interesting to compare your networks with ShuffleNet style networks designed for computational efficiency, rather than networks designed mainly to push the benchmark numbers down whatever the cost.
>
> R)We compared now against: mobileNet, ShuffleNet, HENet, SqueezeNet, we have less number of parameters or better accuracy or both. For instance our method has 4X les parameters than shuffleNet and better accuracy (Added to the paper)
>
> It would be helpful to have the computational cost of the network in FLOPs, and running time compared a regular ConvNet using Winograd/Fourier convolutions.
>
> R)In this paper we focus on the reduction of parameters, we didn’t focus on the speed, we notice that in our experiment our models were trained using half of the epoch used for the conventional models.
> In terms of the number of operations the LeNet as in the tutorial has 2.29M MAC operations, while our method has 1.23M MAC operations for MNIST. (Added to the paper)

---

### Official Review · AnonReviewer2 · 2018-11-03
**Review of "Adaptive Convolutional Neural Networks"**

**Rating:** 5
**Confidence:** 3

**Review:**

This paper presents a pretty cool idea for enabling "adaptive" kernels for CNNs which allow dramatic reduction in the size of models with moderate to large performance drops.  In at least one case, the training time is also significantly reduced (2x).

The best part about this paper is that the size of the models are much smaller; but the paper does offer any explanation of the value of this.  For example, even a 1% drop in accuracy can be unacceptable; but in some applications (like cell phones and IOT devices) model size is critical.  The authors' should add some wording to explain this value.

The "adaptive"kernels the the authors talk about are really a new class of nonlinear kernels.  It would be very interesting to see a discussion of the class of functions these nonlinear kernels represent.  This kind of discussion would give the reader  motivation for the choice of function, ideas for how to improve in this class of functions, and insight into why it works.

The method presented is interesting; but it is not clear that it is present with enough detail for it's results to be replicated.  It would be nice if the authors pointed to a git repository with their code an experiments.  More importantly, the results presented are quite meager.  If this is a method for image recognition, it would be better to present results for a more substantial image recognition problem than MNIST and CIFAR-10.  And the analysis of the "dynamic range" of the algorithim is missing.  How do performance and model size trade off?  How were the number of layers and kernels chosen?  Was the 5x10x20x10 topology used for MNIST the only topology tried?  That would be very surprising.  What is the performance on all of the other topologies tried for the proposed algorithm?  Was crossvalidation used to select the topology?  If so, what was the methodology.

Additionally, some readers may find this paper a little difficult to read due to (1) lack of clarity in the writing, e.g., the first three paragraphs in Section 3; (2) omitted details, e.g., how much overlap exists between kernels (Figs. 1, 2, and 4 suggests there is no overlap - this should be made clear); and (3) poor grammar and nonstandard terminology, e.g., the authors' use of the word "energy" and the phrase "degradation problem".  All of these issues should be addressed in a future version of the paper.

Not sure why Eqns. 2 and 9 need any parentheses.  They should be removed.

---

> ### Author Response · Authors · 2018-11-21
> **Thank you for your very good feedback**
>
> The best part about this paper is that the size of the models are much smaller; but the paper does offer any explanation of the value of this.  For example, even a 1% drop in accuracy can be unacceptable; but in some applications (like cell phones and IOT devices) model size is critical.  The authors' should add some wording to explain this value.
>
> R) Thank you this good observation (Added to the paper)
>
> The "adaptive"kernels the the authors talk about are really a new class of nonlinear kernels.  It would be very interesting to see a discussion of the class of functions these nonlinear kernels represent.  This kind of discussion would give the reader  motivation for the choice of function, ideas for how to improve in this class of functions, and insight into why it works.
>
> The method presented is interesting; but it is not clear that it is present with enough detail for it's results to be replicated.  It would be nice if the authors pointed to a git repository with their code an experiments.
>
> R) Now we have a pytorch version of the code at https://github.com/adapconv/adaptive-cnn
> With MNIST and CIFAR.
>
>
> More importantly, the results presented are quite meager.  If this is a method for image recognition,
> 1)	it would be better to present results for a more substantial image recognition problem than MNIST and CIFAR-10.
>
> R) We added a new experiment for a real life application; testing different topologies.
>
> 2)	And the analysis of the "dynamic range" of the algorithim is missing.
> R) New data was added to the paper exercising multiple topologies, in a wider range of applications.
>
> 3) How do performance and model size trade off?
> R)     A new experiment added to the paper shows the accuracy degradation vs model compression
>
> 4) How were the number of layers and kernels chosen?
>
> R)We started with the original topology replacing convolutional kernels by the Adaptive kernels, then we reduced kernel by kernel, retraining the model each time to match the accuracy (with small drop). But our proposal is not the topology is the new type of filters, so many topologies can be improved using this type of filters, for instance an Adaptive ResNet.
>
> 5) Was the 5x10x20x10 topology used for MNIST the only topology tried?
> R)We tested many, and we think that we can continue reducing the model, but our purpose is not to present a topology, our purpose was to show the advantages of Adaptive convolutions, having a model 66X smaller, 2X less MAC operations and trained 2X faster give us the clue that many researchers can explore on their own topologies and get benefits of it.
>
>   6) That would be very surprising.  What is the performance on all of the other topologies tried for the proposed algorithm?
> R) A table comparing different topologies is included in the new version of the paper.
>
> 7) Was crossvalidation used to select the topology?  If so, what was the methodology.
> We started with the reference topology like: ResNet18, LeNet, etc. then we reduce the number of kernels and layers keeping similar accuracy.
>
> Additionally, some readers may find this paper a little difficult to read due to (1) lack of clarity in the writing, e.g., the first three paragraphs in Section 3; (2) omitted details, e.g.,
> 1)how much overlap exists between kernels (Figs. 1, 2, and 4 suggests there is no overlap - this should be made clear);
> R)That is right, there is not overlap.
>
> and (3) poor grammar and nonstandard terminology, e.g., the authors' use of the word "energy" and the phrase "degradation problem".  All of these issues should be addressed in a future version of the paper.
>
> R) Terms like “energy”  were removed from the paper. We didn’t invent the terminology “degradation problem” it was used here https://arxiv.org/pdf/1512.03385.pdf, you want us to remove it?
> (Fixed on the paper)
>
> Not sure why Eqns. 2 and 9 need any parentheses.  They should be removed. (Fixed on the paper)

---

### Meta-Review · Area_Chair1 · 2018-12-13
**evaluation and results not convincing**

**Confidence:** 5
**Recommendation:** Reject

**Metareview:**

The paper presents a modification of the convolution layer, where the convolution weights are generated by another convolution operation. While this is an interesting idea, all reviewers felt that the evaluation and results are not particularly convincing, and the paper is not ready for acceptance.